# Hydroxyl super rotors from vacuum ultraviolet photodissociation of water

Yao Chang[1,2], Yong Yu[1,3], Heilong Wang[1,4], Xixi Hu [5], Qinming Li[1,3], Jiayue Yang[1], Shu Su[1], Zhigang He[1], Zhichao Chen[1], Li Che[4], Xingan Wang[2], Weiqing Zhang[1], Guorong Wu [1], Daiqian Xie [5], Michael N.R. Ashfold [6], Kaijun Yuan[1] & Xueming Yang[1,7]

Hydroxyl radicals (OH) play a central role in the interstellar medium. Here, we observe highly rotationally excited OH radicals with energies above the bond dissociation energy, termed OH "super rotors", from the vacuum ultraviolet photodissociation of water. The most highly excited OH($X$) super rotors identified at 115.2 nm photolysis have an internal energy of 4.86 eV. A striking enhancement in the yield of vibrationally-excited OH super rotors is detected when exciting the bending vibration of the water molecule. Theoretical analysis shows that bending excitation enhances the probability of non-adiabatic coupling between the $\tilde{B}$ and $\tilde{X}$ states of water at collinear O–H–H geometries following fast internal conversion from the initially excited $\tilde{D}$ state. The present study illustrates a route to produce extremely rotationally excited OH($X$) radicals from vacuum ultraviolet water photolysis, which may be related to the production of the highly rotationally excited OH($X$) radicals observed in the interstellar medium.

[1] State Key Laboratory of Molecular Reaction Dynamics, Dalian Institute of Chemical Physics, Chinese Academy of Sciences, 457 Zhongshan Road, 116023 Dalian, China. [2] Department of Chemical Physics, School of Chemistry and Materials Science, University of Science and Technology of China, 230026 Hefei, Anhui, People's Republic of China. [3] University of Chinese Academy of Sciences, 100049 Beijing, People's Republic of China. [4] Department of Physics, School of Science, Dalian Maritime University, 1 Linghai Road, 116026 Dalian, Liaoning, People's Republic of China. [5] Key Laboratory of Mesoscopic Chemistry, School of Chemistry and Chemical Engineering, Institute of Theoretical and Computational Chemistry, Nanjing University, 210093 Nanjing, People's Republic of China. [6] School of Chemistry, University of Bristol, Bristol BS8 1TS, UK. [7] Department of Chemistry, Southern University of Science and Technology, 518055 Shenzhen, People's Republic of China. These authors contributed equally: Yao Chang, Yong Yu, Heilong Wang. Correspondence and requests for materials should be addressed to X.H. (email: xxhu@nju.edu.cn) or to K.Y. (email: kjyuan@dicp.ac.cn) or to X.Y. (email: xmyang@dicp.ac.cn)

Hydroxyl (OH) is a key radical in interstellar oxygen chemistry, because of its ability to react with most gases in the interstellar medium (ISM)[1,2]. OH radical production routes in the ISM include the dissociative recombination of electrons and molecular cations formed by ion-neutral reactions (e.g., in cold interstellar clouds), photodesorption from the surface of icy grains and, at much higher temperatures, gas-phase atom-molecule collisions, for example, O + H$_2$ and H + H$_2$O[3].

Photodissociation of H$_2$O—in the gas phase or in the mantle of ice grains—is another potential OH radical source. Photoexcitation by the interstellar radiation field will favor the first absorption band (see below) and yield ground state OH($X^2\Pi$) radicals with little rotational excitation. But OH($X$) radicals with extraordinarily high levels of excitation (with energies up to 2.43 eV, corresponding to a rotational quantum number $N = 34$ and an excitation temperature $\approx$28,200 K) have been observed in emission from, for example, HH 211[4] (one of the youngest known stellar outflows) and the T Tauri star DG Tau[5]. Such rotationally "hot" OH radicals are very unlikely to be formed via any of the foregoing chemical reactions, but could originate from short wavelength photolysis of H$_2$O. Clearly, an in-depth understanding of the OH product quantum state population distributions arising in the photodissociation of H$_2$O is an essential prerequisite for any such chemistry modeling.

The photodissociation of H$_2$O has been the subject of many prior experimental and theoretical studies[6–10]. The electronic absorption spectrum of water is concentrated in the vacuum ultraviolet (VUV) region, at wavelengths, $\lambda$, shorter than 190 nm[11]. Excitation in the range 150 < $\lambda$ < 190 nm populates the first excited singlet $(\tilde{A}\,^1B_1)$ state, resulting in direct dissociation to yield an H atom and an OH ($X^2\Pi$) radical with only modest rotational and vibrational excitation[12–15]. The photodissociation of H$_2$O in its second $(\tilde{B}\,^1A_1 - \tilde{X}\,^1A_1)$ absorption band centered at $\lambda$~128 nm, in contrast, serves as a prototype for exploring and understanding non-adiabatic dynamics in polyatomic systems[16–20]. Direct dissociation following excitation to this state yields electronically excited OH $(A^2\Sigma^+)$ products, together with an H atom. However, the major dissociation process from this state yields ground state OH($X$) products via non-adiabatic transitions at the conical intersections (CIs) between the $\tilde{B}$ and $\tilde{X}$ state potential energy surfaces (PESs) at linear H–O–H and O–H–H geometries[21–24]. Further, the OH($X$) rotational state distributions following photoexcitation of H$_2$O at the Lyman-α wavelength ($\lambda = 121.57$ nm)—which is abundant in stellar radiation—show a striking odd-even intensity alternation that has been attributed to dynamical interference between pathways that pass through the rival CIs[25–27]. So far, however, there are few experimental studies that address the dissociation of H$_2$O following excitation to higher electronic states because of the lack of intense VUV lasers in the wavelength region shorter than the Lyman-α wavelength.

The development of the intense, pulsed VUV free electron laser (FEL) at the Dalian Coherent Light Source offers a tool for state-of-the-art molecular photodissociation dynamics studies in the entire VUV wavelength region. The H$_2$O absorption spectrum shows a progression of features in the range 115 < $\lambda$ < 125 nm attributable to population of different ($v_1$ $v_2$ $v_3$) vibrational levels of the $\tilde{C}$ and $\tilde{D}$ Rydberg states (where $v_1$, $v_2$, and $v_3$ represent the symmetric stretch, bend, and anti-symmetric stretch modes, respectively)[28]. Photodissociation dynamics of H$_2$O on these states are believed to relate with the production of highly rotationally excited OH($X$) radicals observed in the ISM, and can now be studied by FEL radiation.

Here, we compare and contrast the fragmentation dynamics when exciting to three different vibrational levels of the $\tilde{D}\,^1A_1$ state of H$_2$O—the (000), (100), and (110) levels, populated by

excitation at, respectively, 121.57, 117.5, and 115.2 nm (Supplementary Figs. 1, 2, and Supplementary Note 1). State-resolved photofragment translational energy spectra recorded using the high-resolution H atom Rydberg tagging technique show that bending vibrational excitation in the $\tilde{D}$ state encourages a much enhanced yield of vibrationally excited OH($X$) super rotors, and that such striking dissociation dynamics only occur at wavelengths ~115.2 nm.

## Results

**Product translational energy distributions**. H atom time-of-flight (TOF) spectra were recorded following photolysis of a jet-cooled H$_2$O sample at $\lambda = 121.57$, 117.5, and 115.2 nm and converted to the corresponding H atom translational energy distributions. Applying momentum conservation arguments yields the total translational energy distribution,

$$E_T = \frac{1}{2} m_H \left(\frac{d}{t}\right)^2 (1 + m_H/m_{OH}), \qquad (1)$$

where $m$ is the mass of the photofragment, $d$ is the flight length from the interaction region to the detector, and $t$ is the H atom TOF measured over this distance. Figure 1 shows the $E_T$ distributions from H$_2$O photodissociation at 117.5 and 115.2 nm using two different polarization schemes. The spectrum at 121.57 nm (Lyman-α) is identical to that recorded previously using a table-top VUV source from four-wave mixing[27] and is shown in the Supplementary Figs. 3 and 4 (Supplementary Note 2). At all three wavelengths, the spectra obtained with the polarization vector (ε) of the VUV-FEL radiation aligned perpendicular to the

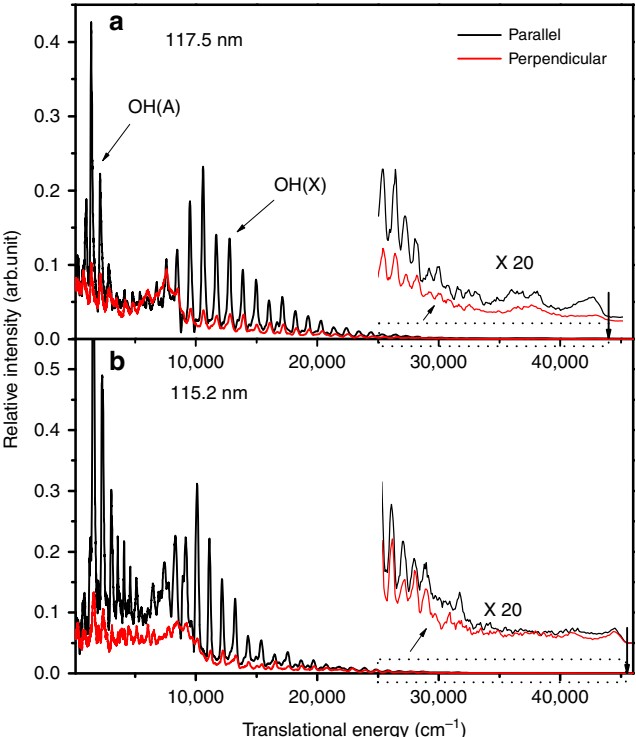

**Fig. 1** Translational energy spectra from H$_2$O photodissociation. Translational energy spectra derived from H atom time-of-flight (TOF) spectra following photodissociation of H$_2$O at 117.5 nm (**a**) and 115.2 nm (**b**), with the detection axis parallel (black) or perpendicular (red) to the photolysis laser polarization, $\varepsilon_{phot}$. The inset displays the spectra in the high translational energy region scaled by a factor of 20

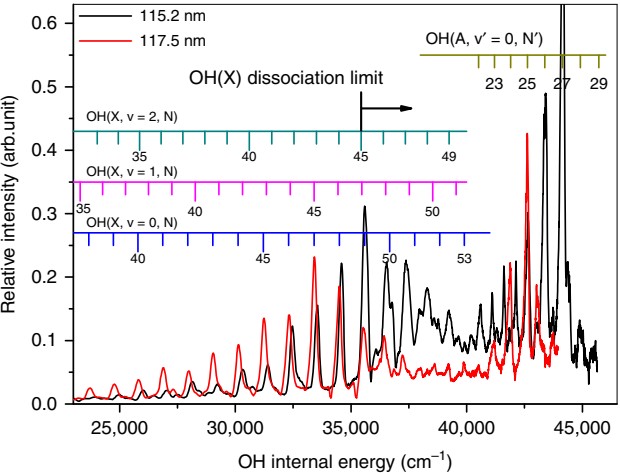

**Fig. 2** Internal energy spectra of OH from $H_2O$ photodissociation. Internal energy spectra of the OH products from photodissociation of $H_2O$ at 117.5 nm (red) and 115.2 nm (black), obtained from H atom time-of-flight (TOF) spectra recorded with the detection axis parallel to the photolysis laser polarization. The sharp features can all be assigned to population of rovibrational states of OH(X) and OH(A). The OH(X) bond dissociation energy, $D_0(O–H)$, is also indicated

detection axis have similar profiles to, but are much weaker than, those recorded with **ε** aligned parallel to the detection axis, consistent with photolysis involving parallel (i.e., $\tilde{D}^1A_1 \leftarrow \tilde{X}^1A_1$) excitation and subsequent prompt dissociation.

The internal energies of the OH fragments ($E_{int}(OH)$) formed with the detected H atoms can be deduced from conservation of energy arguments. Energy conservation requires that, for the unimolecular dissociation $H_2O + h\nu \rightarrow H + OH$ ($\alpha$, $v$, $N$) (where $\alpha$, $v$, $N$ represent the electronic, vibrational, and rotational states of the OH product),

$$E_{int}(H_2O) + E_{h\nu} - D_0(H−OH) = E_{int}(OH) + E_T, \quad (2)$$

where $E_{int}(H_2O)$ is the internal energy of the parent $H_2O$ molecule, $E_{h\nu}$ is the photon energy, and $D_0(H−OH)$ is the dissociation energy[29]. In the supersonic beam, the rotational temperature of $H_2O$ molecule is estimated to be ~10 K[25]. Figure 2 shows the internal energy distributions of the OH fragments formed in the photodissociation of $H_2O$ at 115.2 and 117.5 nm. Based on previous spectroscopic data[30], all of the intense sharp features in the spectrum obtained at $\lambda = 117.5$ nm can be readily assigned to high rotational levels of OH(X, $v = 0$) and OH(A, $v = 0$) radicals. Vibrationally excited OH(X) products are also observable in this spectrum, but with much lower relative yields. This energy disposal is similar to that observed when exciting $H_2O$ at the Lyman-α wavelength[27]. The spectrum obtained at $\lambda = 115.2$ nm shows noticeable differences, however. Careful inspection of the $25000 < E_{int}(OH) < 40,000$ cm$^{-1}$ region shows obvious peak shifts compared to those measured at 117.5 nm or at 121.57 nm; the dominant peaks in this region of the $\lambda = 115.2$ nm spectrum indicate population of high rotational levels of OH(X) products with $v > 0$. The peaks at $E_{int}(OH) > 40,000$ cm$^{-1}$ are still attributable to high rotational levels of OH(A, $v = 0$) products, however.

**Observation of OH super rotors.** One of the most striking findings from the $\lambda = 115.2$ nm photolysis of $H_2O$ is the formation of extremely rotationally excited OH fragments. The most intense peak in the $E_{int}(OH)$ spectrum involves contributions from the $v = 0$, $N = 49$, $v = 1$, $N = 47$, and $v = 2$, $N = 45$ levels of OH(X), with internal energies ~4.41 eV, that is, above the bond dissociation energy of the OH radical ($D_0(O–H) = 4.39$ eV[27]). We

term such extremely rotationally excited OH(X) products "super rotors," and note that vibrationally excited super rotors are even more pronounced (see Fig. 2). The highest energy peak unambiguously associated with OH(X) in Fig. 2 can be assigned to $v = 2$, $N = 49$ products. These have an energy $E_{int}(OH) = 4.86$ eV (equivalent to a wavenumber of 39,235 cm$^{-1}$ and an excitation temperature ≈56,490 K), almost 0.5 eV above the dissociation limit. We note that even higher energy levels of OH(X) (e.g., the $v = 2$, $N = 50$, or $v = 1$, $N = 51–53$ levels) may well be populated also, but that it is hard to determine the population of these states with certainty given the extent to which they are overlapped by OH(A) products. The super rotors observed here have never been observed previously in the photodissociation of $H_2O$, though Harich et al.[31] identified (less highly excited) OH(X) super rotors in the $\lambda = 121.57$ nm photolysis of HOD. Figure 3 shows the OH(X, $v = 0–4$) rotational state population distributions obtained by simulating the $E_{int}(OH)$ spectra obtained from $H_2O$ photolysis at $\lambda = 115.2$ nm. The level corresponding to the OH bond dissociation limit in each panel is marked by a red arrow. This analysis suggests that >30% of the OH(X) products have energies lying above $D_0(O–H)$, and that most of these super rotors are vibrationally excited.

The key to understanding the observed rotational energy disposal is the remarkable topography of the PESs of water. Van Harrevelt and van Hemert[18] identified a strong interaction between the $\tilde{D}$ and $\tilde{B}$ states of $H_2O$ attributable to an avoiding crossing at bent geometries. The photodissociation of $\tilde{D}$ state $H_2O$ molecules starts with fast non-adiabatic conversion to the $\tilde{B}$ state PES; the subsequent fragmentation dynamics are controlled by the topography of the latter PES. Figure 4 shows contour plots of the $\tilde{B}$ and $\tilde{X}$ state PESs constructed by one of us[20] with one OH bond length fixed at 1.07 Å (the mean bond length for centrifugally extended OH(X) products with high $N$ values).

OH(X) products form after passage through either of the well-documented H−O−H or O–H–H CIs between the $\tilde{B}$ and $\tilde{X}$ state PESs[18,21,25]. The $\tilde{B}$ state PES displays deep (~2.5 eV) potential wells at collinear HOH and OHH geometries. It is repulsive with respect to O–H bond extension in the Franck–Condon region, but its strong angular anisotropy ensures that the forces actually point near perpendicular to the radial direction. Thus, the H atom accelerates towards linear HOH geometries and gains a large amount of orbital angular momentum (route 1). The OH partner rotates (clockwise in Fig. 4) in order to compensate for the change in angular momentum of the H atom, thereby ensuring conservation of the total angular momentum. After non-adiabatic coupling to the $\tilde{X}$ state PES, the longer OH bond fully dissociates yielding, predominantly, highly rotationally excited OH (X, $v = 0$) products.

In addition, a fraction of the $H_2O$ ($\tilde{B}$) population overcomes the barrier at HOH bond angles $\gamma \approx 60°$[18] and propagates towards the O–H–H CI (route 2 in Fig. 4). This CI lies above the H−O−H CI in energy, and the dynamics in this region are complicated by the nature of the PESs at short H−H separations[27]. Dissociating molecules become temporarily trapped and can make multiple jumps between the $\tilde{B}$ and $\tilde{X}$ state PESs, facilitating intramolecular vibrational redistribution between H−H and O−H vibrational motions that manifests itself as a wider range of OH(X) product vibrational excitation[27]. In general, the dissociative flux arising via route 2 is a relatively minor fraction of the total yield and H + OH (X, $v = 0$) products dominate—as observed at $\lambda = 117.5$ and 121.57 nm.

**Enhanced production of vibrationally excited OH super rotors.** The simulations of the $E_{int}(OH)$ spectra return the OH(X) and

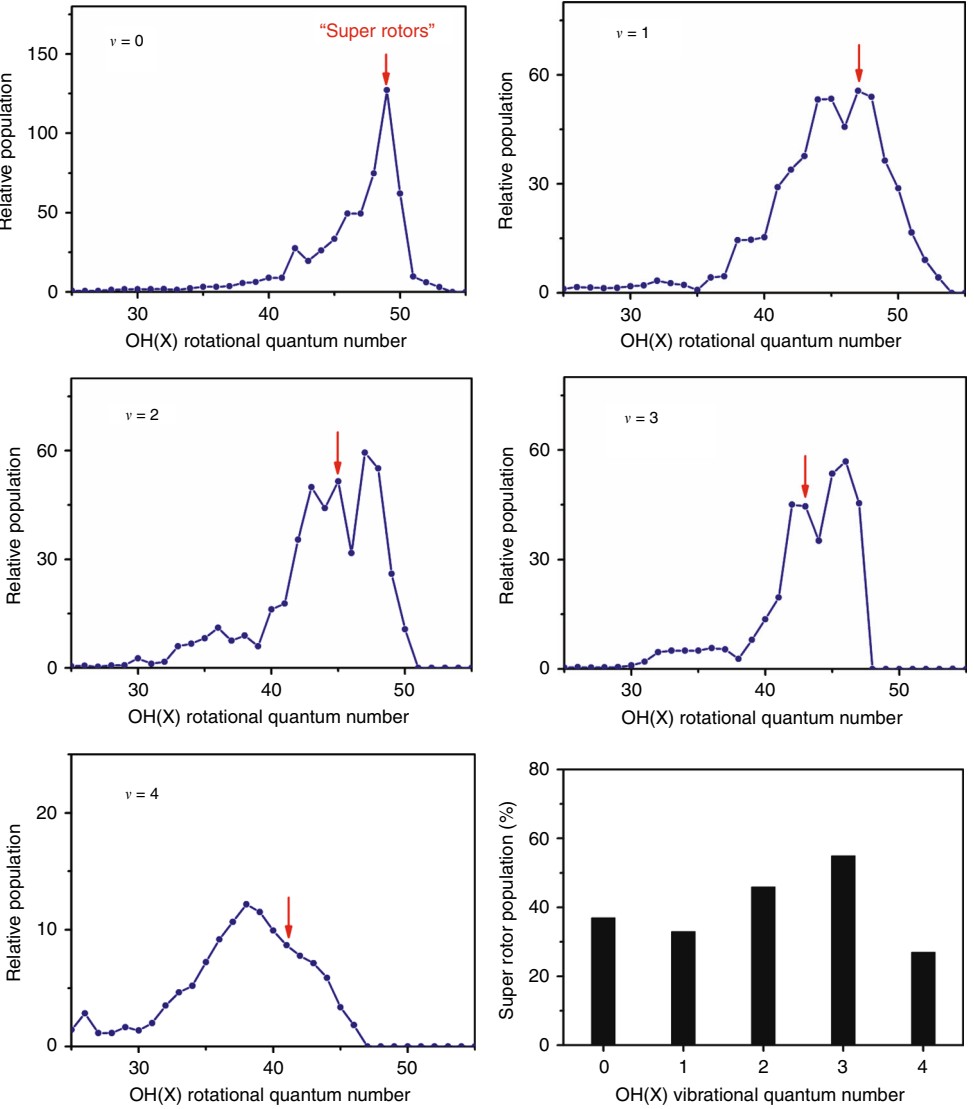

**Fig. 3** Rotational state distributions of OH from $H_2O$ photodissociation. Rotational state population distributions of the OH($X$, $\nu = 0-4$) products formed in the photodissociation of $H_2O$ at 115.2 nm, and the super rotor population as a function of vibrational quantum number. The first super rotor level, above the O ($^3$P) + H dissociation limit, is indicated in each rotational population panel by the red arrows. The higher super rotor levels lie to the right of the red arrows

OH($A$) vibrational state population distributions shown in Fig. 5. The vibrational excitation of the OH($X$) products formed by photolysis at $\lambda = 115.2$ nm is clearly much higher than in those formed at $\lambda = 117.5$ or 121.57 nm. As noted above, excitation at $\lambda = 115.2$, 117.5, and 121.57 nm populates, respectively, the (110), (100), and (000) levels of the $\tilde{D}\,^1A_1$ state of $H_2O$. Non-adiabatic coupling from the $\tilde{D}$ to $\tilde{B}$ state is efficient, so nuclear motions introduced in the photoexcitation step will be conserved upon radiationless transfer to the $\tilde{B}$ state PES. The topography of the $\tilde{B}$ state PES ensures that the subsequent nuclear acceleration is concentrated in the bending coordinate, towards linear geometries (Fig. 4). Activating the bending coordinate (e.g., by exciting the parent (110) level, with $\nu_2 > 0$) can enhance the fraction of the dissociative flux propagating towards the O–H–H CI and thereby boost the population of vibrationally excited OH($X$) products, especially the vibrationally excited OH super rotors. Inspection of Fig. 5 shows that ~73% of the OH($X$) products formed when exciting to the $\tilde{D}$(110) state are in levels with $\nu > 0$ (cf. ~30% when exciting the $\tilde{D}$(000) state). The symmetric stretch vibration, in contrast, has little effect on the OH($X$) vibrational state

population distribution; the relative yields of OH($X$, $\nu > 0$) products when exciting the $\tilde{D}$(100) and $\tilde{D}$(000) states are very similar. Such observations accord with the preceding discussion. The symmetric stretch acts at right angles to the bending coordinate and thus has little influence on the way flux branches towards the two CIs on the $\tilde{B}$ state PES. The similarities between the OH ($X$, $\nu = 0$) rotational population distributions observed following excitation to the $\tilde{D}$(100) and $\tilde{D}$(000) states (peaking at $N = 47$ and $N = 45$, respectively) suggest that the quantum of symmetric stretch vibration simply partitions into the total energy available to the dissociating $H_2O$ molecules.

The OH($A$) vibrational state population distributions are also shown in Fig. 5. The OH($A$) products are formed by following the adiabatic pathway, that is, by dissociating on the $\tilde{B}$ state PES, and their vibrational state distributions are only weakly dependent on any initial vibrational excitation of the $H_2O$ parent.

The OH($X$) super rotors, with an energy above the dissociation limit, are only stable by virtue of the associated centrifugal barriers, that is, by the additional contribution to the potential energy of the form $N(N + 1)h^2/8\pi^2\mu R^2$, where $N$ is the rotational

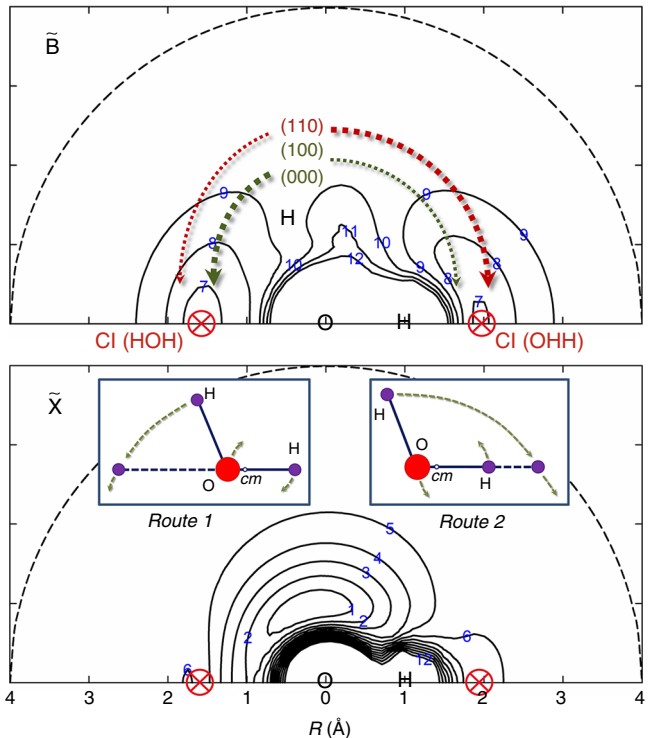

**Fig. 4** Potential energy surface contour plots of $H_2O$. Contour plots of the $\tilde{B}$ and $\tilde{X}$ state potential energy surfaces (PESs) for the motion of H around OH with a fixed OH bond length of 1.07 Å. Energies are given in eV relative to the minimum of the ground state. The conical intersections (CIs) at the linear H−O−H and O−H−H geometries, where minima of the $\tilde{B}$ state PESs are degenerate with maxima of the $\tilde{X}$ state PES, are shown in red. The $\tilde{B}$ and $\tilde{X}$ state PESs correlate adiabatically with, respectively, H + OH($A$) and H + OH($X$) products. The dotted curves in the upper panel depict the dissociative flux through the two CIs, with a thicker pen thickness implying relatively larger yield. The inset panels show the "route 1" and "route 2" pathways through the respective CIs

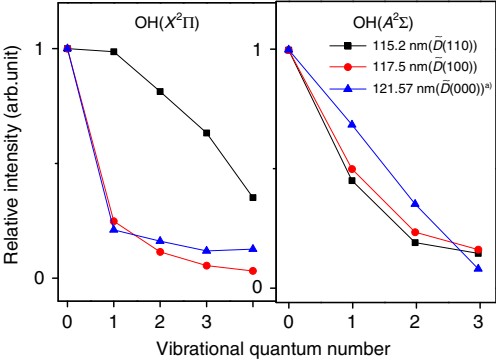

**Fig. 5** Vibrational state distributions of OH from $H_2O$ photodissociation. Vibrational state distributions of the OH($X$) and OH($A$) products from the photodissociation of $H_2O$ at $\lambda = 115.2$, 117.5, and 121.57 nm, corresponding to excitation of the $\tilde{D}(110)$, $\tilde{D}(100)$, and $\tilde{D}(000)$ states, respectively. (The data for the $\tilde{D}(000)$ state are adapted from ref. [27])

quantum number, $R$ is the internuclear distance, and $\mu$ is the reduced mass of OH. Radicals formed in these super-rotationally excited levels can dissociate by tunneling through the centrifugal barrier. Tunneling lifetimes for highly rotationally excited levels of OH($X$, $v = 0$) were estimated by Harich et al.[31] Based on that study, we have also estimated tunneling lifetimes of the super-rotationally excited levels of OH($X$, $v > 0$) (Supplementary

Note 3). As Supplementary Fig. 5 shows, the tunneling lifetimes drop steeply with increasing $v$. By way of illustration, the tunneling lifetime of the $N = 50$ level of the $v = 0$ state of OH($X$) was estimated to be much longer than the age of the universe ($13.4 \times 10^9$ years)[32], whereas the estimated tunneling lifetimes of the $N = 50$ levels of the $v = 1$, 2, and 3 states are, respectively, a few hours, microseconds, and <100 ps. The highest energy OH($X$) level unambiguously identified in the present work (Fig. 2) has $v = 2$ and $N = 49$ (each of the higher levels shown as having small populations in Fig. 3 are blended in the total kinetic energy release spectra), the estimated lifetime of which is a few milliseconds. This suggests that most of the super rotors identified in the present work are metastable and might have a role in OH related chemical reactions. The super rotors have significantly stretched bond lengths, and thus might be expected to show unusual collisional properties. We also note that the high yields of vibrationally excited OH super rotors from $H_2O$ are limited to photolysis wavelengths $\lambda \sim 115.2$ nm. Such products are not observed in the photodissociation of $H_2O$ at even shorter photolysis wavelengths[33,34], thus highlighting the peculiar non-adiabatic dissociation dynamics that prevail in this narrow wavelength range.

**Implications for super excited OH rotors in interstellar chemistry**. Since the OH radical is an important driver of chemistry in the ISM, it is clearly prudent to explore all possible sources of OH radicals for modeling such environments. Infrared (IR) emission studies identified an extraordinary sequence of lines in HH 211 attributable to high rotational levels of OH($X$) with energies up to 2.43 eV above the ground state (excitation temperature $\approx 28,200$ K)[4,5]. Similar emissions have been identified in the mid-IR spectrum of the T Tauri star DG Tau[35]. Tappe et al.[36] cautioned that the populated levels of OH($X$) in the latter environment might well extend to yet higher energies, but that detector limitations prevented them from observing emission from such levels. They also suggested that excited state photo-chemistry of $H_2O$ in the 115–140 nm absorption band might be responsible for the observed rotationally excited OH radicals, recognizing that possible alternative OH($X$) formation processes like $H_2O$ photolysis in its first ($\tilde{A} - \tilde{X}$) absorption band, or photodesorption from the icy surface of interstellar grains, or the aforementioned bimolecular reactions could not account for the observed emissions from such high rotational levels. That being so, the observation of emissions characteristic of such highly rotationally excited OH($X$) radicals can act as a proxy for the presence of water in different regions of the ISM.

The absorption cross-section of $H_2O$ at 115.2 nm is $\sigma_{H_2O} \sim 5 \times 10^{-18}$ cm$^2$ (ref. [37]). Given the previously reported (small) cross-section for non-H atom-forming channels[38], the translational energy resolved differential cross-sections for forming H atoms can be derived, as shown in Supplementary Fig. 6. Branching ratios for the H + OH($\alpha$, $v$, $N$) and 2H + O($^3$P) product channels were then determined via spectral simulations (Supplementary Fig. 7, and Supplementary Note 4), enabling estimation of the cross-section for forming OH($X$) super rotors at $\lambda = 115.2$ nm: $\sigma \sim 6.3 \times 10^{-19}$ cm$^2$. The abundance of VUV photons in interstellar space suggests that the contributions of these OH sources could be significant and thus should be recognized in appropriate interstellar chemistry models.

**Discussion**
The present study provides a benchmark illustration of the use of a FEL-based VUV light source to exploit non-adiabatic couplings between electronic states of $H_2O$ and thereby demonstrates the production of vibrationally excited OH super rotors. These super

rotors account for >30% of the total OH(X) product yield at the photolysis wavelength $\lambda \sim 115.2$ nm.

## Methods

**Vacuum ultraviolet free electron laser radiation**. The experiment employs a recently constructed instrument for molecular photochemistry centered on the VUV-FEL beam line[34,39]. Briefly, the VUV-FEL facility operates in the high gain harmonic generation mode[40], in which the seed laser is injected to interact with the electron beam in the modulator. The seeding pulse within the wavelength range $240 < \lambda_{seed} < 360$ nm is generated from a Ti:sapphire laser. The electron beam is generated from a photocathode radio frequency gun, and accelerated to a beam energy of ~300 MeV by seven S-band accelerator structures, with a bunch charge of 500pC. The micro-bunched beam is sent through the radiator, which is tuned to the $n$th harmonic of the seed wavelength, and coherent FEL radiation with wavelength $\lambda_{seed}/n$ is emitted. Optimization of the linear accelerator yields a high-quality beam with emittance ~1.5 mm·mrad, projected energy spread of ~1‰, and pulse duration of ~1.5 ps. The VUV-FEL presently operates at 10 Hz, the maximum pulse energy is ~500 μJ/pulse and the output wavelength is continuously tunable across the range 50–150 nm. The typical spectrum bandwidth of the VUV-FEL output is ~50 cm$^{-1}$.

**The H atom Rydberg tagging time-of-flight spectroscopy**. The H atom Rydberg tagging TOF technique used in this work was pioneered by Welge and coworkers[41]. The cornerstone of this technique is the sequential two-step excitation of the H atom photofragments. This involves initial excitation of the H atom on the $n = 2 \leftarrow n = 1$ transition by absorption of a 121.57 nm photon, followed by UV laser excitation (365 nm) from the $n = 2$ level to a high-$n$ Rydberg state. The neutral Rydberg atoms then fly about 280 mm to reach a rotatable microchannel plate (MCP) detector where they are ionized by the electric field, giving a TOF temporal resolution <0.5%. The sample beam was generated by expanding a mixture of $H_2O$ and Ar at a stagnation pressure of 600 Torr through a 0.5 mm diameter pulsed nozzle. The VUV-FEL light beam crosses the molecular beam at right angles. Since the polarization of the VUV-FEL light pulse is fixed in the horizontal plane, the "parallel" and "perpendicular" photodissociation spectra are obtained by rotating the MCP detector around the propagation axis of the VUV-FEL output. The 121.57 nm detection laser beam also generates a background H atom spectrum, which is obtained by turning the VUV-FEL beam on and off, and subtracted accordingly.

## Data availability

The data supporting this study are available from the authors on reasonable request.

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

## Acknowledgements

The experimental work is supported by the Strategic Priority Research Program of the Chinese Academy of Sciences (Grant No. XDB 17000000), the Chemical Dynamics Research Center (Grant No. 21688102), the National Natural Science Foundation of China (NSFC Nos. 21873099, 21673232, and 21673234), and the Youth Innovation Promotion Association (2014160). The theoretical work is supported by NSFC (Nos. 21733006 and 21590802). MNRA gratefully acknowledges funding from the Engineering and Physical Sciences Research Council (EPSRC, EP/L005913) and the NFSC Center for Chemical Dynamics for the award of a Visiting Fellowship. This paper is dedicated to the 70th anniversary of Dalian Institute of Chemical Physics, Chinese Academy of Sciences.

## Author contributions

K.Y. and X.Y. designed the experiments. Y.C., K.Y., Y.Y., H.W., Q.L., J.Y., S.S., Z.H. and Z.C. performed the experiments. K.Y., Y.C., H.W. and S.S. analyzed the data. X.H. and D.X. performed the tunneling lifetime calculations and provided the PES figure. K.Y., M.N.R.A., X.Y., L.C., X.W., W.Z. and G.W. discussed the experimental results. K.Y., M.N.R.A. and X.Y. prepared and revised the manuscripts.

## Additional information

**Competing interests:** The authors declare no competing interests.

