## [Peer Review File · Nature Communications]

Review of “Vibrationally Excited OH Super Rotors from Vacuum Ultraviolet Photodissociation of H₂O,” by Chang, ... Ashfold, Yuan, and Yang, submitted to Nature Communications.

This manuscript is a clearly written account of an important experiment in which it is shown that hydroxyl radicals produced in the 115-nm photodissociation of water are extremely rotationally excited, even to energies above the barrier for dissociation for a rotationless OH radical. The finding is important because 1) it is an unusual example of production of such high rotation, 2) the dynamical processes producing such high rotation need to be understood, and 3) it may be important to understanding OH emission from the interstellar medium. The figures are excellent and the references are appropriate and comprehensive. I urge publication, essentially in its current form. There are a few minor typographical problems that can be fixed in the proof stage (e.g., references not superscripted on page 12).

Reviewer #2 (Remarks to the Author):

The work provides a clear experimental evidence for the formation of OH super-rotors in photodissociation of H₂O at 115 nm. The observed super-rotors possess rotational energy higher than the dissociation energy of the OH radical. The results are relevant to astrochemical observations of spectra of highly rotationally excited OH radicals in the ISM. The experiments take advantage of the novel experimental technique - free electron laser producing highly intense short-wave-length VUV light recently commissioned in Dalian. The paper also provides clear theoretical explanation of the observed results based on the available excited potential energy surfaces of H₂O and the structure of conical intersections. The manuscript is very nicely and logically written and, in my opinion, deserves publication in Nature Communication. In the meantime there are a couple points for the authors to address:

- 1) In the emission from ISM, the observed highly rotationally excited OH radicals have energies up to 2.43 eV, while the super-rotors observed in this experiment have an internal energy of 4.68 eV. How this apparent inconsistency can be explained?
- 2) The statement in Abstract concerning the enhancement of the non-adiabatic transition between the B and X states of H₂O does not reflect the full picture described in the paper because the excitation initially occurs to the D state. It would be useful if the authors clarify this statement and talk about D to B transition first, followed by the B to X transition.

Vibrationally Excited OH Super Rotors from Vacuum Ultraviolet Photodissociation of H₂O

This manuscript describes the observation of vibrationally excited OH “super rotors”, *i.e.*, extremely highly rotationally excited OH radicals with energies greater than the bond dissociation energy, in the vacuum ultraviolet (VUV) photodissociation of H₂O using a newly developed tunable VUV free electron laser. The yield of vibrationally excited OH super rotors is enhanced when the bending vibration of the parent H₂O molecule is excited. Theoretical analysis provides insight into the photodissociation mechanism: \tilde{D} state H₂O molecules starts with fast non-adiabatic conversion to the \tilde{B} state PES; the subsequent fragmentation dynamics are controlled by the topography of the \tilde{B} state PES and the conical intersections (CIs) of the \tilde{B} and \tilde{X} state PESs. The authors also argue that the present study provides an explanation for the observed emissions from very highly rotationally excited OH(X) radicals in the ISM.

The observation of the super rotor OH(X) is striking, and the mechanism by itself is also interesting. But the connection of the super rotor OH(X) (such as $v=0$, $N=49$) to highly rotationally excited OH(X) radicals observed in ISM seems to be tentative and might be a little bit of stretch, as the highly rotationally excited OH(X) radicals in ISM have rotational quantum numbers up to $N = 34$ and they could come from VUV photodissociation of H₂O without the production of the super rotor OH(X) radicals. This consideration could reduce the impact of this manuscript.

Additional points:

- (1) Both the title and the abstract (and the text) term vibrationally excited OH super rotors. But ground vibrational state OH super rotors were also observed.
- (2) P5, lines 102-110, and SI, lines 74-96, the sharp peaks at 121.57, 117.5 and 115.2 nm could be assigned to the ($v_1 v_2 v_3$) vibrational levels of either the \tilde{C} or \tilde{D} Rydberg states. But it is argued that they are more likely to be assigned to the (000), (100) and (110) levels of the \tilde{D} state. Is there any information on the vibrational frequencies of the \tilde{D} state? Any H-atom product action spectra near these three peaks? Furthermore, the (010) level of the \tilde{D} state is expected to be near 119.1 nm, likely the peak between (000) at 121.57 nm and (100) at 117.5 nm. Was photodissociation experiment carried out near 119 nm? This could reveal the effect of a pure bending vibrational excitation in the \tilde{D} state.
- (3) For eqn 2, the rotational temperature of H₂O in the beam should be specified.
- (4) Page 7, some more details need be provided (in SI) on how the simulation was carried out to extract the rotational populations of OH(X) $v=0,1,2,3,4$, particularly for 115.2 nm. The peaks (e.g., in OH internal energy 30000-37500) are not obviously broadened by the overlapping rotational levels of $v = 0-4$. For example, the peak at ~ 34500 cm⁻¹ could have contributions from $v=0$, $N=48$; $v=1$, $N=46$; $v=2$, $N=44$; and perhaps $v=2$, $N=42$; and $v=4$, $N=40$. It is not clear how the relative contributions of these states were determined in the simulation. Also in Figure 3, the rotational population of $v=4$ should be plotted.

- (5) Page 7, lines 155-157, it is stated that “The highest energy peak clearly associated with OH(X) in Fig. 2 can be assigned to $v=2$, $N=49$ products”. But as shown in Figure 3, $v=1$, $N=51-53$ are present, which are of higher internal energy than $v=2$, $N=49$.
- (6) Figure S4 is not mentioned in the main text or in SI.
- (7) SI should have page numbers.
- (8) Page 10, lines 220-224, the $\tilde{D}(100)$ distribution peaks at $N=47$ (not 48).
- (9) Page 11, lines 241-242, the same as point (5), the statement is not consistent with Figure 3 ($v=1$, $N=51-53$ are present, which are of higher internal energy than $v=2$, $N=49$).
- (10) Page 11, lines 242-251, the lifetimes of the vibrationally excited super rotor states seem to be rather short in comparison with the likely collision time scale in ISM. They may not play a significant role in collisions, despite their unusual collision property.
- (11) Page 12, lines 270-277, and SI, lines 146-183, the calculations of the cross sections need be revised. As obtained from the simulation in Figure 7, the total H-atom yield ratios of H + OH(X): H+OH(A): O(³P)+2H are 0.39:0.26:0.35. The branching ratios of these three channels should be 0.39:0.26:0.18 (1/2 of the H-atom yield for the O(³P)+2H channel). The estimated cross section for forming OH(X) super rotors at $\lambda = 115.2$ nm would be $\sigma \sim 6.3 \times 10^{-19}$ cm².
- (12) Page 13, line 284, need clarify the role of OH radical in the interstellar oxygen chemistry.
- (13) Figure 1, in the 117.2 nm and 115.2 nm photodissociation of H₂O, there might be some H-atom signals from the 121.57 nm photodissociation of H₂O by the Lyman-alpha probe laser. How much is this contribution? Was it removed from the spectra in Figure 1 and others?
- (14) SI, line 54, the acronym HRTOF need be defined.

Author's responses to reviewers' comments

Reviewer #1

This manuscript is a clearly written account of an important experiment in which it is shown that hydroxyl radicals produced in the 115-nm photodissociation of water are extremely rotationally excited, even to energies above the barrier for dissociation for a rotationless OH radical. The finding is important because 1) it is an unusual example of production of such high rotation, 2) the dynamical processes producing such high rotation need to be understood, and 3) it may be important to understanding OH emission from the interstellar medium. The figures are excellent and the references are appropriate and comprehensive. I urge publication, essentially in its current form. There are a few minor typographical problems that can be fixed in the proof stage (e.g., references not superscripted on page 12).

Author Reply: Thank you very much for your comments. We have fixed the typographical problems in the text.

Reviewer #2

The work provides a clear experimental evidence for the formation of OH super-rotors in photodissociation of H₂O at 115 nm. The observed super-rotors possess rotational energy higher than the dissociation energy of the OH radical. The results are relevant to astrochemical observations of spectra of highly rotationally excited OH radicals in the ISM. The experiments take advantage of the novel experimental technique - free electron laser producing highly intense short-wave-length VUV light recently commissioned in Dalian. The paper also provides clear theoretical explanation of the observed results based on the available excited potential energy surfaces of H₂O and the structure of conical intersections. The manuscript is very nicely and logically written and, in my opinion, deserves publication in Nature Communication. In the meantime there are a couple points for the authors to address:

1) In the emission from ISM, the observed highly rotationally excited OH radicals have energies up to 2.43 eV, while the super-rotors observed in this experiment have an internal energy of 4.68 eV. How this apparent inconsistency can be explained?

Author Reply: Thank you very much for your comments. The key reason why the ISM emission studies only reveal rotationally excited OH(X) radicals with energies ≤ 2.43 eV is that the wavelength of the emission from the ($N = 34$) rotational level (*i.e.* of the $N = 34 \rightarrow N = 33$ transition) is ~ 10.06 μm and this is the short wavelength detection limit of the short-high (SH) detector module on the Spitzer telescope. Tappe *et al*, *ApJ* (2008) (ref. 4 in submitted manuscript) specifically comment that ‘the population of the OH energy levels probably extends to even higher energies, but the corresponding rotational transitions fall in the wavelength range of the low-resolution IRS modules below 10 μm ’. This is borne out by inspecting fig. 2 in that paper or, better still, the expanded version of the spectrum shown in fig. 2 of Tappe *et al.*, *ApJ* **751**, 9 (2012)) (We have added this reference as Ref. 36 in the revised manuscript) which suggests similar intensities for lines involving all levels in the range $N = 19$ to 34. There is no indication that the $N = 34$ level is the highest populated level of OH in these regions of the ISM – this ‘cut-off’ is simply a reflection of the detector response.

No mechanism other than VUV photolysis of H₂O has been advanced as a source of such highly rotationally excited OH radicals in the ISM (see Carr *et al*, *ApJ* **788**, 66 (2014), ref. 5 in submitted manuscript). But, as Carr *et al.* point out, even with an optimized detector capable of detecting the necessary shorter wavelengths, the analysis of spectra of ISM emissions should not be expected to return such high levels of product rotational excitation as found in photofragment translational spectroscopy experiments such as those reported in the present manuscript. The astrophysical OH spectra do not sample the instantaneous distribution in which OH is formed, as the present experiments are designed to do. Estimates of the Einstein A coefficients at $N = 45$ are as high as ~ 600 s⁻¹, and hence the most highly rotationally excited OH radicals will rapidly decay down the rotational ladder.

2) The statement in Abstract concerning the enhancement of the non-adiabatic transition between the B and X states of H₂O does not reflect the full picture described in the paper because the excitation initially occurs to the D state. It would be useful if the authors clarify this statement and talk about D to B transition first, followed by the B to X transition.

Author Reply: Thank you very much for your suggestions. We have added this statement to the Abstract. Please see Page 2, line 15-16.

Reviewer #3

This manuscript describes the observation of vibrationally excited OH “super rotors”, *i.e.*, extremely highly rotationally excited OH radicals with energies greater than the bond dissociation energy, in the vacuum ultraviolet (VUV) photodissociation of H₂O using a newly developed tunable VUV free electron laser. The yield of vibrationally excited OH super rotors is enhanced when the bending vibration of the parent H₂O molecule is excited. Theoretical analysis provides insight into the photodissociation mechanism: D state H₂O molecules starts with fast non-adiabatic conversion to the B state PES; the subsequent fragmentation dynamics are controlled by the topography of the B state PES and the conical intersections (CIs) of the B and X state PESs. The authors also argue that the present study provides an explanation for the observed emissions from very highly rotationally excited OH(X) radicals in the ISM.

The observation of the super rotor OH(X) is striking, and the mechanism by itself is also interesting. But the connection of the super rotor OH(X) (such as $v=0$, $N=49$) to highly rotationally excited OH(X) radicals observed in ISM seems to be tentative and might be a little bit of stretch, as the highly rotationally excited OH(X) radicals in ISM have rotational quantum numbers up to $N = 34$ and they could come from VUV photodissociation of H₂O without the production of the super rotor OH(X) radicals. This consideration could reduce the impact of this manuscript.

Author Reply: Thank you very much for your comments. We agree that the levels of OH rotational excitation (up $N=34$) seen in the ISM emission are quite lower in energy than that measured in our lab experiments. This is mainly due to the short wavelength detection limit of the short-high (SH) detector module on the Spitzer telescope, which cannot observe higher OH rotational levels via short wavelength radiation. Therefore the OH rotational excitation might be much higher than $N=34$ in the ISM. We have explained this in the response to reviewer #2, question #1. Also as noted there, there is no conceived mechanism other than vacuum ultraviolet (VUV) photodissociation of H₂O capable of explaining the emission from such highly rotationally excited OH radicals in the relevant regions of the ISM. Given the dynamics picture in the VUV H₂O photodissociation, VUV photolysis seems to be the only likely source of OH(X, high N). The mechanism to produce extremely rotationally excited OH via the B-X conical intersection in the current work is therefore related to that of the observed OH high rotationally excited states in the ISM.

Additional points:

(1) Both the title and the abstract (and the text) term vibrationally excited OH super rotors. But ground vibrational state OH super rotors were also observed.

Author Reply: Thank you very much for your suggestions. We have revised the term “vibrationally excited OH super rotors” to “OH super rotors” in the title, and in several places in the abstract and the text.

(2) P5, lines 102-110, and SI, lines 74-96, the sharp peaks at 121.57, 117.5 and 115.2 nm could be assigned to the ($v_1 v_2 v_3$) vibrational levels of either the C or D Rydberg states. But it is argued that they are more likely to be assigned to the (000), (100) and (110) levels of the D state. Is there any information on the vibrational frequencies of the D state? Any H-atom product action spectra near these three peaks? Furthermore, the (010) level of the D state is expected to be near 119.1 nm, likely the peak between (000) at 121.57 nm and (100) at 117.5 nm. Was photodissociation experiment carried out near 119 nm? This could reveal the effect of a pure bending vibrational excitation in the D state.

Author Reply: Thank you very much for your comments. The room temperature absorption spectrum of H₂O has been reported by several groups including Bell (ref. 28 in the submitted manuscript) and Mota *et al.* (ref. 11 in the submitted manuscript). The 80000-90000 cm⁻¹ energy range of current interest has also been investigated quite extensively by 3+1 REMPI methods (see *e.g.* Ashfold *et al. Chem. Phys.* **84**, 35 (1984); *Can. J. Phys.* **62**, 1806 (1984)). The REMPI studies reveal predissociated C state levels (and higher Rydberg states); the D state levels predissociate more rapidly (by coupling to the B state) and do not contribute to the REMPI spectra. Prior studies of the C-X transitions show that the 121.57 nm peak is nothing to do with a C-X resonance. Comparing REMPI and absorption data provides some discrimination between C-X and D-X transitions. Bell's β -system corresponds to transitions to vibrational levels of the D¹A₁ state. He reports vibrational intervals ($v_1 = 3258$ cm⁻¹, $v_2 = 1636$ cm⁻¹) to a precision that is not justified by the breadth of the features in his spectrum, but which are nonetheless sensibly consistent with the corresponding ground state values and with expectations based on a Rydberg \leftarrow non-bonding orbital promotion. We are not aware of any H-atom product action spectra in this region, but Suto and Lee (*Chem. Phys.* **110**, 161

(1986), Ref. 37 in the submitted manuscript) have highlighted the similarity of the action spectrum for forming OH (A) products and the parent H₂O absorption spectrum in this wavelength range.

No experiments were undertaken in the present study at $\lambda \sim 119$ nm, such as would excite the D ($v_2=1$) \leftarrow X transition. Given the fundamental wavenumbers $\nu_1=3179$ cm⁻¹, $\nu_2=1407$ cm⁻¹ and $\nu_3=3238$ cm⁻¹ reported for the C state (ref. 28 in the submitted manuscript), absorption to the C (100) state can be predicted to span the wavelength region 119.2 to 119.4 nm (consistent with the REMPI data shown in Ashfold *et al.*, *Chem. Phys.* **84**, 35 (1984)), which partially overlaps with that of the D (010) state (~ 119.5 nm).

(3) For equation 2, the rotational temperature of H₂O in the beam should be specified.

Author Reply: Thank you very much for your suggestions. In the supersonic beam, the rotational temperature of the H₂O molecules in the molecular beam is estimated to be ~ 10 K. We have added this statement in the text. Please see Page 6, lines 14-15.

(4) Page 7, some more details need be provided (in SI) on how the simulation was carried out to extract the rotational populations of OH(X) $v=0,1,2,3,4$, particularly for 115.2 nm. The peaks (e.g., in OH internal energy 30000-37500 cm⁻¹) are not obviously broadened by the overlapping rotational levels of $v=0-4$. For example, the peak at ~ 34500 cm⁻¹ could have contributions from $v=0$, $N=48$; $v=1$, $N=46$; $v=2$, $N=44$; and perhaps $v=2$, $N=42$; and $v=4$, $N=40$. It is not clear how the relative contributions of these states were determined in the simulation. Also in Figure 3, the rotational population of $v=4$ should be plotted.

Author Reply: Thank you very much for your suggestions. The spectral simulation has been performed using software developed in-house (Ref. 27 in the submitted manuscript), in which we use a Gaussian profile to simulate the population of each quantum state. The width of the profile for each quantum state used in the simulation varies according to the relative kinetic energy resolution ($\delta E/E < 1\%$) and the FEL beam width. (For example, a width of 150 cm⁻¹ has been used in the simulation at TKER ~ 10000 cm⁻¹, increasing to 250 cm⁻¹ at TKER ~ 20000 cm⁻¹). The intensities of each OH quantum state and of the broad feature at low translational energy (attributable to the triple fragmentation to O (³P) + 2 H products) are then adjusted to produce a summed simulated TKER spectrum that should match the experimental

data. This fitting process has been illustrated previously (Ref. 27 in the submitted manuscript). Clearly, this process calls for some judgement on the part of the fitter, since overlapping rotational levels from different vibrational states can obscure the relative contributions of the different levels in the simulation. Normally, we try to ensure that the populations of rotational levels associated with each vibrational state change smoothly, since there is no obvious reason why the rotational populations should fluctuate wildly with small changes in N . We have added this discussion in the SI and added the population of $v=4$ in Figure 3 of the manuscript as suggested.

(5) Page 7, lines 155-157, it is stated that “The highest energy peak clearly associated with OH(X) in Fig. 2 can be assigned to $v=2$, $N=49$ products”. But as shown in Figure 3, $v=1$, $N=51-53$ are present, which are of higher internal energy than $v=2$, $N=49$.

Author Reply: Thank you very much for your comments. The peaks associated with the OH(X, $v=1$, $N=51-53$ levels are totally overlapped with those of OH (A, v , N) levels, and the populations of such states obtained from the fitting are necessarily less certain. The highest energy peak that we can unambiguously associate with OH(X) in Fig. 2 is assigned to $v=2$, $N=49$ products. We have sought to clarify this point in the text so as to remove any potential confusion. Please see Page 7, line 15 and lines 18-21.

(6) Figure S4 is not mentioned in the main text or in SI.

Author Reply: Thank you very much for pointing this out! We have added the description of Figure S4 (Supplementary Figure 4) on Page 10 (Supplementary note 2) of the SI.

(7) SI should have page numbers.

Author Reply: We agree, and have now numbered the pages in the SI.

(8) Page 10, lines 220-224, the D (100) distribution peaks at $N=47$ (not 48).

Author Reply: We agree, the D (100) population distribution does indeed peak at $N=47$ and we have revised this value in the text.

(9) Page 11, lines 241-242, the same as point (5), the statement is not consistent with Figure 3 ($v=1$, $N=51-53$ are present, which are of higher internal energy than $v=2$, $N=49$).

Author Reply: This is a similar point to that raised in (5) above and we have again revised the relevant part of the text to remove any confusion. Please see Page 11, lines 16-18.

(10) Page 11, lines 242-251, the lifetimes of the vibrationally excited super rotor states seem to be rather short in comparison with the likely collision time scale in ISM. They may not play a significant role in collisions, despite their unusual collision property.

Author Reply: Thank you very much for your comments. The estimated lifetimes quoted in the text are specifically for the different OH (v , $N=50$) levels. But fig. 3 shows that all OH ($v=2$, $N > 45$ levels), for example, satisfy our definition of being super rotors. These lower N levels will have much longer lifetimes with respect to tunnelling than the OH ($v=2$, $N=50$) level. The pressure in planetary atmospheres may be quite high – the pressure of Venus’ atmosphere, for example, is ~ 92 times higher than that at the Earth’s surface – and the OH super rotors might play a role in chemical reactions in such environments. However, we also agree that the high level OH super rotors are not likely to play a significant role in collisions in the dilute atmosphere prevailing in the ISM and have revised the description accordingly and removed this particular reference to the ISM.

(11) Page 12, lines 270-277, and SI, lines 146-183, the calculations of the cross sections need be revised. As obtained from the simulation in Figure 7, the total H-atom yield ratios of H + OH(X): H+OH(A): O(3P)+2H are 0.39:0.26:0.35. The branching ratios of these three channels should be 0.39:0.26:0.18 (1/2 of the H-atom yield for the O(3P)+2H channel). The estimated cross section for forming OH(X) super rotors at $\lambda = 115.2$ nm would be $\sigma \sim 6.3 \times 10^{-19}$ cm 2 .

Author Reply: We are most grateful to the reviewer for spotting this error, agree that the estimated cross section for forming OH(X) super rotors at $\lambda = 115.2$ nm would be $\sigma \sim 6.3 \times 10^{-19}$ cm 2 , and have amended this value appropriately where it appears in the text and in the SI.

(12) Page 13, line 284, need clarify the role of OH radical in the interstellar oxygen chemistry.

Author Reply: Thank you very much for your comments. The OH radical is important in interstellar oxygen chemistry in many diverse environments since it participates in many chemical reactions, *e.g.*, $\text{OH} + \text{H}_2 \rightarrow \text{H}_2\text{O} + \text{H}$, $\text{H}_2 + \text{O} \rightarrow \text{OH} + \text{H}$ (*ApJ*, **680**, 117 (2008)) or $\text{H}_2\text{O} + \text{O}(^1\text{D}) \rightarrow \text{OH} + \text{OH}$, $\text{OH} + \text{O} \rightarrow \text{O}_2 + \text{H}$, $\text{OH} + \text{O}_3 \rightarrow \text{HO}_2 + \text{O}_2$ (*Geophys. Res. Letts.*, **25**, 3935 (1998)), *etc.* Given the statement in (and response to) question (10), however, we agree with the reviewer that the OH super rotors may well not have a significant reactive role in the dilute atmosphere of the ISM and have thus removed the relevant words from the text.

(13) Figure 1, in the 117.2 nm and 115.2 nm photodissociation of H_2O , there might be some H-atom signals from the 121.57 nm photodissociation of H_2O by the Lyman-alpha probe laser. How much is this contribution? Was it removed from the spectra in Figure 1 and others?

Author Reply: Thank you very much for this comment and the questions. Indeed, there is some probe laser induced H atom signal, *i.e.* signal due to the 121.57 nm probe laser induced photodissociation of H_2O . The intensity of this signal in the present experiments is about one fifth that induced by the 117.5 and 115.2 nm VUV-FEL pulses. The 121.57 nm background signal is obtained by turning the FEL VUV beam on and off, and subtracted accordingly. Figure 1 (below) illustrates the pump-probe and probe only signals, and we now include this detail in the methods in the main text. Please see Page 15, lines 2-4 in the main text.

Figure 1. The TOF spectra of H atoms from H₂O photodissociation at 117.5 nm (black) and the 121.57 nm probe laser induced background (red), recorded with the 117.5 nm VUV-FEL beam respectively on and off.

(14) SI, line 54, the acronym HRTOF need be defined.

Author Reply: We have added the definition of HRTOF at the relevant point in the methods in the main text and in the SI.

REVIEWERS' COMMENTS:

Reviewer #2 (Remarks to the Author):

The authors have very carefully addressed the comments and suggestions of all three reviewers and I recommend publication of this manuscript in its present form.

Reviewer #2 (Remarks to the Author):

The authors have very carefully addressed the comments and suggestions of all three reviewers and I recommend publication of this manuscript in its present form.

Author reply: Thank you very much for your comments. We are very glad to hear that the reviewer 2 finds our manuscript publishable.